# Early Results of Kinesio Taping and Steroid Injections in Elbow Lateral Epicondylitis: A Randomized, Controlled Study

**DOI:** 10.3390/medicina57040306

**Published:** 2021-03-24

**Authors:** Firat Erpala, Tahir Ozturk, Eyup Cagatay Zengin, Ugur Bakir

**Affiliations:** 1Department of Orthopaedics and Traumatology, Cesme Alpercizgenakat State Hospital, 35930 Cesme/Izmir, Turkey; 2Department of Orthopaedics and Traumatology, Gaziosmanpasa University, 60250 Merkez/Tokat, Turkey; ozturk.tahir@yahoo.com (T.O.); zengincagatay@hotmail.com (E.C.Z.); 3Department of Orthopaedics and Traumatology, Bunyan State Hospital, 38600 Bunyan/Kayseri, Turkey; uur_bakr@yahoo.com

**Keywords:** lateral epicondylitis, Kinesio tape, corticosteroid, tennis elbow

## Abstract

*Background and objectives:* This study prospectively compares early results of Kinesio tape (KT) as an alternative method for the treatment of lateral epicondylitis with those of corticosteroid injection and the rest-and-medication group (RMG). *Materials and methods:* Among the fifty patients (53 elbows), KT was applied to 20 patients (21 elbows), and corticosteroid injection (CSI) was applied to 15 patients (17 elbows). Fifteen patients were included in the RMG. Patients in the RMG were informed about their condition, and necessary warnings were given. No oral or topical treatments were recommended. If needed, paracetamol or nonsteroidal anti-inflammatory drugs (NSAIDs) were prescribed. Patients were classified according to the Nirschl scores and evaluated with visual analog scale (VAS); Quick Disability of Arm, Shoulder and Hand (QDASH); and the Turkish version of the Patient Related Elbow Evaluation (PREE-T). *Results:* Improvements in all scores were statistically significant in all groups at the end of the second week. At the end of the fourth week, there was also a statistically significant improvement in all three groups, but these improvements were not as high as they were in the first 2 weeks. There was a slight deterioration in the functional scores in the RMG and CSI groups, while the improvement in the KT group continued. In the KT group, the average QDASH score was 18.1 (4.5–35), the VAS score was 2 (1–3), the VAS score in resisted wrist extension was 4 (2–5) and the Nirschl score was 2 (1–3) at the fourth week. The KT group had significant superiority in these parameters over the RMG (*p* = 0.035, *p* = 0.035, *p* = 0.029, *p* = 0.035, respectively). However, there was no significant difference between the KT, the RMG and the CSI groups at the fourth week. *Conclusions*: CSI, KT and rest-and-medication treatments were all effective in terms of pain reduction and functional scores at the end of week 2, and the only treatment that continued to be effective in the final week was KT.

## 1. Introduction

Lateral epicondylitis (LE) is a chronic and degenerative disease that reduces a person’s quality of life and activity level [1]. Although the disease affects up to 3% of the population, this rises to 15% in heavy industrial workers [2,3]. LE arises from the excessive and repetitive use of the forearm extensors and is characterized by pain and tenderness in the lateral side of the elbow [2]. It is seen equally in females and males and peaks between the ages of 34 and 55 [2,3,4,5].

There are different treatment methods for LE, both conservative and surgical. Conservative treatments include oral–topical anti-inflammatory drugs, steroid injections, immobilization, extracorporeal shock wave therapy, electrotherapy, laser application, hot–cold application and exercise therapy [6]. In most patients, complaints decrease with these treatment methods, and surgical intervention is required in a small number of patients. Because there are not enough studies reported in the literature about which of the conservative treatment methods is superior, the gold standard treatment protocol has still not been determined [1,2,3,4,5,6,7].

Kinesio tape (KT) is a material that was developed by Kenzo Kase in the late 1970s. It is made of cotton and imitates the skin with a similar thickness and structure that can extend longitudinally 30–40% [8]. The tape is water-resistant and can stay on the skin for up to 7 days [1,9]. The tape reduces pain and increases sensitivity by increasing lymphatic drainage and blood flow in the subcutaneous tissue [10]. It also blocks the pain signals according to the gate control theory by increasing the stimulations to the afferent nervous system [11]. It also has effects in form of muscle stimulation or inhibition depending on the technique of the application [12]. KT is used for therapeutic purposes in various clinical situations (myofascial pain syndrome, subacromial impingement, hemiplegia, lymphedema, tendinitis and patellofemoral pain syndrome). In the literature, there are studies reporting successful results with KT in the treatment of elbow LE [1,12].

Corticosteroid injection (CSI) is a frequently preferred method of treating LE. In the literature, studies report its superiority over other conservative treatments, but there are also studies reporting that its effect is short-term, and the rate of recurrence after treatment is high [13,14,15]. Side effects such as osteomyelitis, cellulitis, ecchymosis, subcutaneous adipose tissue atrophy and hypopigmentation may be observed after the injection [16].

As it is known, many treatment modalities have been tried on LE, but no consensus has been reached [6]. In terms of its easy application and practicality, KT is gaining popularity day by day in the treatment of musculoskeletal disorders. Considering the scarcity of publications in the literature, this study prospectively compares early results of KT as an alternative method for the treatment of LE with those of an established method (CSI) and the rest-and-medication group (RMG).

### Ethical Evaluation

Informed consent was obtained from the patients before the procedures began. The study was conducted in accordance with the Helsinki Declaration, and approval was obtained from the local ethics committee (IRB number: 72 11.07.2019). All patients included in the study were informed about both treatments, and possible side effects were explained. All patients completed informed consent forms.

## 2. Materials and Methods

The patients were admitted to the orthopedics and traumatology clinics between 2018 and 2019 and were diagnosed with LE. They were evaluated randomly.

Patients aged 18–70 years who tested positively in at least one of the Mills, Maudsley or Cozen’s tests and experienced symptomatic pain for at least 6 weeks lateral to the elbow were included in the study. Patients with cervical radiculopathy, neuropathy, a previous diagnosis of rheumatological disease, autoimmune disease, diabetic neuropathy, surgical histories, open wounds or scars, local or systemic infections, pregnancy, histories of allergic reactions to tape or corticosteroid injections or elbow joint arthrosis and patients who had CSI in the elbow area in the last 6 months were not included in the study.

The sample size was based on a significant difference from a previous study [4]. The alpha level was determined as 0.05, the power as 90% and the confidence interval as 95%. The current study was determined to include at least 43 patients. It was designed to include at least 15 patients in three groups.

Out of the 81 patients diagnosed with LE, 6 patients did not meet the inclusion criteria, 7 did not sign the informed consent form and 18 did not attend follow-ups. Fifty patients meeting the inclusion criteria were evaluated. In the KT group, one patient had bilateral LE; in the CSI group, two patients had bilateral LE. Therefore, 53 elbows were evaluated for the final outcome. For randomization purposes, patients who applied to the clinic on Monday received KT, and those who applied on Wednesday received the CSI. The patients who applied on Thursday formed the RMG. KT was applied to 20 patients meeting the inclusion criteria, and CSI was applied to 15 patients. Fifteen patients were included in the RMG (shown in Figure 1).

### 2.1. Kinesio Tape Application

The Kinesio tape was applied to the patients in the KT group by two certified authors (FE and UB) using muscle and area correction techniques as described by Kase et al. [8].

In each application, the Kinesio tape was kept for 5 days, and a total of three applications were performed. Patients were not allowed to take the tape off, and on the fifth day of each application, the tape was taken off by researchers. Before each new application was administered, the skin was cleaned and dried.

In the muscle technique, taping was applied on two legs along the direction of the extensor carpi radialis brevis (ECR) and extensor digitorum communis (EDC) with a strain starting from the distal insertion of the ECR to its origin at the elbow level. At the distal and proximal parts of the tape, the anchor portions of 5 cm were applied without tension. The second tape was adhered to the lateral epicondyle over the muscle origins with full tension, except for the 5-cm anchor portions (shown in Figure 2). The patients were advised to avoid contact with water as much as possible, and they underwent three sessions of KT every 5 days. During the treatment, no measurements were taken from the patients. They were evaluated after the KT treatment ended.

### 2.2. Corticosteroid Injection

In the CSI group, 10 mg triamcinolone (20 mg/mL 0.5 cc) without dilution by any agent was used. The CSI was applied in the first examination after the functional and pain evaluations. Under sterile conditions, after proper staining, a single injection with a 22-G 30 mm needle was applied by the same authors (FE and UB) to the most painful point in the origin of the ECR on the lateral epicondyle.

### 2.3. Rest-and-Medication Group

Patients in the RMG were informed about their condition, and necessary warnings were given. No oral or topical treatment (analgesic and anti-inflammatory) or physiotherapy was recommended. Patients were told that the disease may regress only by waiting and with daily activity modifications. Paracetamol (2000–4000 mg 110 daily) or nonsteroidal anti-inflammatory drugs (NSAIDs) (naproxen 1000 mg daily) were prescribed to all patients in the group for use if needed during the first 2 weeks of the treatment period.

During the study, all patients were instructed to avoid activities that could intensely irritate the elbow, such as lifting heavy objects, washing laundry, squeezing, kneading dough, using small hand tools such as drill-pliers or screwdrivers and gardening. No specific exercise was recommended for any patient in any of the groups. During the study, patients were not allowed to use oral/topical NSAID drugs outside the RMG. All patients completed the study.

### 2.4. Outcome Measures

All participants were evaluated during the first application, at week 2 and at week 4 by a researcher. In the evaluation, patients were classified according to the Nirsch LE scoring scale, which consists of seven stages (Table 1). The severity of pain on the lateral epicondyle at rest and the resistance to wrist extensor muscle activity was assessed by visual analog scale (VAS). Patients were asked to express the pain they felt on the VAS scale as a value between 0 and 10. If there was no pain, the patient chose 0, and 10 meant there was unbearable pain. Quick Disability of Arm, Shoulder and Hand (QDASH) and Turkish version of the Patient-Related Elbow Evaluation (PREE-T) scores were used for functional evaluations.

For the QDASH scores, patients responded to the 11 evaluation questions by marking the appropriate answer on the 5-point Likert scale (1: difficulty; 5: not at all). Each patient was given a score between 0 (best case) and 100 (worst case). For the PREE-T scores, 10 questions and a total of 50 points evaluated patients’ activity and function. The patient answered the questions on a scale of 0 to 10 (0 meant no pain, and 10 meant the most severe pain the patient had experienced). The measurements for grip strength were evaluated with the Jamar brand (model SH 5001, Saehan Corporation Masan, South Korea) hand dynamometer.

As recommended in the literature by some authors, the Jamar hand dynamometer was used, and measurements were made when the shoulder was adducted, the elbow was flexed at 90 degrees and the forearm was in the neutral position, in which grip strength was the most efficient [17]. The mean values of the three consecutive measurements were recorded.

### 2.5. Statistical Analysis

The SPSS 22.0 software (SPSS Inc., Chicago, 103 IL, USA) was used for data analysis. For data distribution, the Kolmogorov–Smirnov test was used. The ANOVA (one-way and repetitive measurements) test was applied for the parameters with normal distribution, and the Kruskal–Wallis test was applied for non-normally distributed data. The Mann–Whitney U test was applied among the groups for non-normally distributed data. The post hoc Tukey analysis was applied to the parameters that were significant with the ANOVA test. Categorical variables were evaluated with chi-square tests, and a *p* value less than 0.05 was considered significant.

## 3. Results

No significant difference was found when age (*p* = 0.427), dominant hand (*p* = 0.238) and affected hand (*p* = 0.875) of the patients in all three groups were compared (Table 2). The dominant hand was affected for 13 patients in the KT group (one patient had bilateral LE), 10 patients in the CSI group (two patients had bilateral LE) and 8 patients in the RMG. There was no statistically significant difference when hand dominance was compared between the groups (*p* = 0.875). The improvement in the average QDASH, PREE-T, VAS, resisted extension VAS (eVAS), Nirschl staging and hand-grip measurements at the end of the second week was statistically significant in all groups (Table 3).

When we compared the improvement between the groups at the end of the second week, the CSI group had statistical superiority over the RMG in all parameters except in the QDASH scores. However, only eVAS scores and PREE-T scores in the CSI group improved significantly compared to those in the KT group (Table 3).

At the end of the second week, there was no significant improvement in the KT group compared to the RMG (Table 3).

At the end of the fourth week, there was a statistically significant improvement in QDASH, PREE-T, VAS, eVAS and Nirschl staging compared to the pre-treatment period in all three groups, but these improvements were not as high as they had been after the first 2 weeks. There was a slight deterioration in the hand-grip measurements in the RMG, while the improvement in the KT and CSI groups continued (Table 3).

The CSI group did not show statistically significant superiority over the KT group in any parameters, but it was superior to RMG in all parameters. The KT group exhibited superiority over RMG in all parameters, except for the PREE-T scores at the end of the fourth week (Table 3).

When pre-treatment hand-grip measurements were evaluated, it was observed that the KT group had lower hand-grip power compared to the other groups. This difference was statistically significant with the RMG (*p* = 0.022), but not significant with the CSI group (*p* = 0.126). There was no significant difference between the CSI group and the RMG in terms of hand-grip measurements (*p* = 0.723). Hand-grip measurements showed a statistically significant improvement in all groups at the end of week 2 (Table 3). In the fourth week, there was a statistically significant increase compared to the pre-treatment values in the KT and CSI groups, but there was no significant increase in the RMG (Table 3).

## 4. Discussion

This study aimed to compare the short-term effectiveness of CSI and KT applications in treating LE. Functional (PREE-T, QDASH, Nirschl staging) scores, VAS scores and hand-grip strength were found to be statistically significantly improved within all groups at the end of the second week. At the end of the fourth week, the CSI and KT groups had superiority over the RMG, but CSI had no advantage over KT in any parameters.

Since CSIs have been used for many years as a method in the treatment of LE, there are many studies in the literature that have focused on this treatment [2,5,13]. Koçak et al. stated that all three treatment methods were effective in terms of pain reduction, functional scores and patient satisfaction at the end of weeks 3 and 12 for 84 patients treated with steroid injections, KT and a combination of both treatments [18]. As in our study, the authors stated that the effectiveness of KT alone was close to that of steroid administration alone. However, their study did not include a control group [18]. Although CSI was superior to all other palliative treatments in terms of pain reduction, functional scores and grip strength in the short-term follow-up (2 to 6 weeks), there are studies reporting that NSAIDs and physiotherapy are superior in medium- and long-term follow-ups [19,20,21].

Although paracetamol and oral NSAIDs are often employed as the first-line treatment of LE, the efficacy of these interventions has not been established and there are contradictory results in the literature regarding these treatments. Hay et al., in their study, compared the daily dose of 1000 mg naproxen with placebo and corticosteroid injections. They found the CSI group to be superior compared to the placebo and naproxen groups [21]. Pattanittum et al. also reported conflicting results on the superiority of oral NSAIDs in their systematic review, and the authors found insufficient evidence of oral NSAIDs being more effective than placebo alone in the short term for reducing pain [22].

In parallel with studies in the literature, our study also showed the CSI group to have superior results compared to the RMG. However, considering the side effect profile and the contradictory results in the literature, we tried to inform our patients in the RMG about the impact of rest and activity modifications on the treatment regimen and explain that medications could be used only when needed.

In this study, which aimed to compare the short-term results, we found CSI was more effective than other treatment methods, especially at the end of the second week, but not superior to KT at the end of the fourth week. A possible reason why CSI was effective in the short-term is that pain in LE is thought to be due to the release of substance P, calcitonin and glutamate from primary sensory nerves [23]. Glutamate is a neurotransmitter responsible for the passage of pain, which is quite abundant in enthesopathies. Corticosteroids may also provide a sudden reduction in pain due to their potential to completely inhibit neurotransmitters and receptors [23]. For this reason, although there is no change in the main pathology in the short-term, the pain may decrease due to the analgesic effect of corticosteroid. A possible cause of CSI ineffectiveness in the long term is the exposure of the injected tissue to a degenerative process rather than inflammatory processes, and high doses of CSI can negatively affect the reparative ability of the tissue, thereby delaying the self-limiting process of the tissue by preventing the release of cytokines [24].

KT has recently become more popular for the treatment of LE. Dilek et al. stated that there was an improvement in pain reduction, functional grip strength, patient satisfaction and disease staging at the end of the second and sixth weeks in their studies with 31 LE patients who were administered KT and who completed home physiotherapy programs. However, they stated that these positive results may have occurred as a result of a placebo effect, as there was no control group in the study [1]. In our study, we did not give any physiotherapy programs to the patients to whom we applied KT, except for activity recommendations. In this way, we aimed to evaluate the impact of KT without including other factors. At the end of the second week, we found KT was not superior to the RMG in terms of pain reduction, functional scores and improvement of grip power. In the fourth week, the efficacy in all scores in the CSI group and the RMG decreased, while this effect persisted in the KT group. In addition to the original placebo effect, we believe that KT reduces pain by helping to reduce irritation in chemical receptors in the area where it is applied, improving circulation and possibly reducing pain with the door control mechanism [3].

To eliminate the placebo effect of taping, placebo-controlled studies have used sham tape and informal tape, reporting different results. In a recent meta-analysis about the effects of KT and sham taping on musculoskeletal conditions, Ramirez-Velez et al. stated that KT had superior effects on pain at follow-up evaluations. However, they also stated that in patients with low back pain, KT did not reveal any improvements in pain reduction in the post-treatment period and in disability scores [25]. Cho et al. stated that KT had better results than sham taping in the treatment of LE, especially in resisted wrist extension [3]. Celik et al. evaluated 14 studies comparing KT and sham taping in shoulder disorders. In their meta-analysis, they concluded that there is no firm evidence of any benefit of KT over sham taping [26]. This study mainly aims to compare early results of KT and CSI with a control group. Therefore, sham taping or informal taping groups were not designed to eliminate the placebo effect of the taping process.

It was observed that the KT group had lower hand-grip measurements in the first application. Even though there was an increase in the grip strength when it was analyzed at the end of the second week within all three groups, it was observed that the KT group had lower results compared to the other groups in terms of hand-grip power. However, there was a slight decrease in the CSI group and RMG at the end of the fourth week, and the improvement in the KT group continued at the end of this week. In our study, although the KT group had less hand-grip strength in the pre-treatment period, the increase of the grip strength was superior compared to the other groups, and at the end of the fourth week in the post-treatment period, there were no statistically significant differences across all groups. There are few studies in the literature about the effect of kinesiological taping on grip strength, and Halseth et al. have shown that KT has positive effects on grip strength and muscle activity, but more comprehensive randomized controlled studies are needed [27,28].

The determination of the actual daily activities of the patients remains uncertain. Besides that, patients in the KT group might have done less daily activity due to the constant feeling of tape on their elbows. However, between the second and fourth weeks, even though patients in the KT group did not have any taping sessions, an improvement was observed in all the parameters evaluated.

There are some limitations in our study. The researchers who made the measurements in the evaluation were not blind to the study because they also planned it. The duration of symptoms was not recorded. Patients with persistent symptoms over longer periods might have affected the outcomes in their groups. Paracetamol or nonsteroidal anti-inflammatory drugs were prescribed to all patients in the RMG during the final 2 weeks of the treatment period for them to use if needed. Patients were advised not to use medications, but they were informed that they could use prescribed medications in painful conditions. This situation creates a bias as the use of medication is left to the will of the patient. Since the patients in the RMG may have used medication of their own free will, we can conclude that this might be the reason that the KT group did not outperform the RMG at the end of the second week. However, despite the bias, the KT group had a significant improvement over the RMG between the second and fourth weeks while patients did not have any treatments in both groups. In accordance with the literature, we believe that KT improved the disease over the long term due to effects such as the regulation of muscle and tendon activity, the alignment of ligaments and the regulation of regional lymphatic drainage [1,3,4,6,8,9,10,11].

In addition, due to the short follow-up period, it was not possible to evaluate the long-term effects in the CSI and KT groups. Other limitations include small patient groups and the diagnosis of LE based only on clinical findings, not diagnostic imaging methods.

## 5. Conclusions

CSI, KT and rest-and-medication treatments were all effective in terms of pain reduction, functional scores and grip strength at the end of week 2, and the only treatment that continued to be effective in the final week was KT. We believe that KT is an alternative treatment option that is as effective as CSI because of the long-term uncertainties of CSI in the treatment of LE.

## Figures and Tables

**Figure 1 medicina-57-00306-f001:**
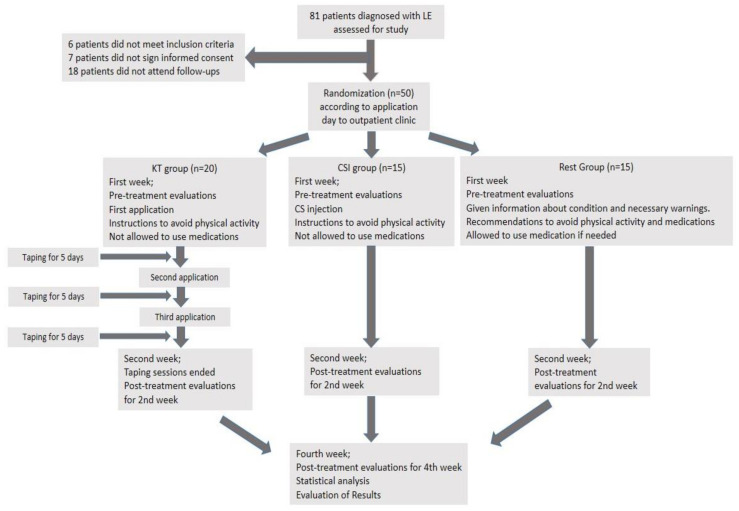
Evaluation of patients.

**Figure 2 medicina-57-00306-f002:**
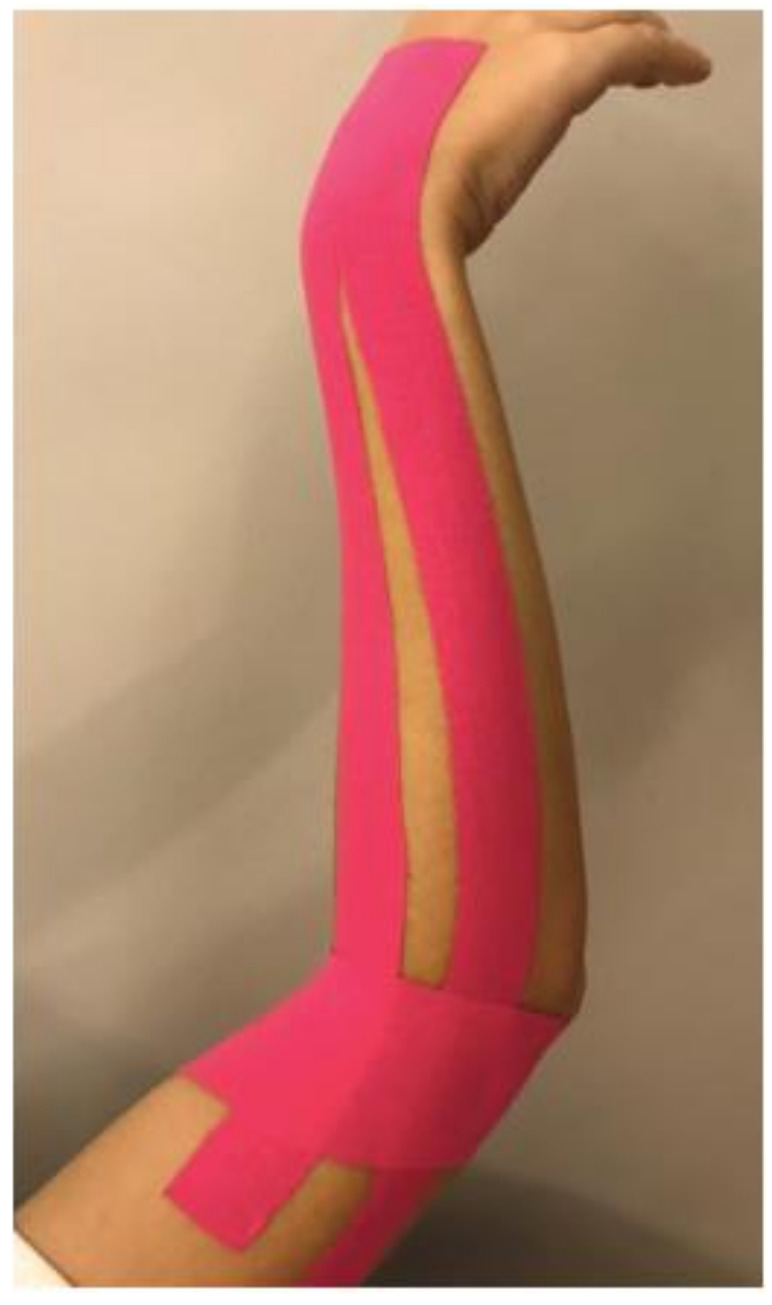
Muscle inhibition Kinesio tape (KT) applied from the origin to the insertion of the extensor muscle group.

**Table 1 medicina-57-00306-t001:** Nirschl lateral epicondylitis pain evaluation scale.

Phase	Clinic
Phase 1	Mild pain after exercise, recovers in 24 h
Phase 2	Pain after exercise, recovers in 48 h
Phase 3	Pain after exercise, does not require altering the activity
Phase 4	Pain after exercise, requires altering the activity
Phase 5	Pain occurring during intense activities of daily living
Phase 6	Pain occurring during simple activities of daily living, pain sometimes present at rest
Phase 7	Constant pain at rest, disrupts sleep

**Table 2 medicina-57-00306-t002:** Demographic characteristics of patients.

	CSI	KT	RMG	Total
**Patient (elbow)**	17	21	15	53
**Age (Mean ± SD)**	47.0 ± 8.5	48.2 ± 9.1	44.5 ± 7.3	46.8 ± 8.4
**Sex (F/M)**	5 (33.3%)/10 (66.7)	3 (15%)/17 (85%)	5 (33.3%)/10 (66.7%)	13/37
**Profession (High hand power/low hand power/unemployed)**	8/4/3	8/6/6	7/5/3	23/15/12
**Dominant hand (right/left)**	12 (80%)/3 (20%)	17 (85%)/3 (15%)	13 (84.7%)/2 (15.3%)	42/8
**Affected hand (right/left)**	9 (52.9%)/8 (47.1%)	16 (77.2%)/5 (23.8%)	8 (53.4%)/7 (46.6%)	33/20
**VAS Pre-treatment (Median IQR)**	8 (7–8)	8 (7–9)	7 (6–8)	8 (7–8)
**QDASH Pre-treatment (Mean ± SD)**	58.3 ± 13.6	58.8 ± 12.7	50.8 ± 14.1	56.4 ± 13.6
**Nirschl Score (Median IQR)**	6 (5–6)	6 (5–7)	5 (4.5–6)	6 (5–7)
**Hand Grip Pre-treatment (Mean ± SD)**	25.5 ± 11.3	19.4 ± 8.3	28.1 ± 8.6	23.8 ± 10.0

**Table 3 medicina-57-00306-t003:** Clinical findings and *p* values of comparisons in repeated measures and between groups.

	CSI	KT	RMG	CSI vs. KT	CSI vs. RMG	KT vs. RMG
QDASH						
Pre-treatment	58.3 ± 13.6	58.8 ± 12.7	50.8 ± 14.1	0.990	0.273	0.194
Post-treatment 2nd week	22.5 ± 15.4 **< 0.001 ^a^**	30.9 ± 15.4 **< 0.001 ^a^**	31.4 ± 14.2 **< 0.001 ^a^**	0.217	0.234	0.995
Post-treatment 4th week	12.5 (4.5–47.5) **< 0.001 ^b^**	18.1 (4.5–35) **< 0.001 ^b^**	40.8 ± 14.0 **< 0.001 ^b^**	0.929	**0.039**	**0.035**
PREE-T						
Pre-treatment	71.5 (64.5–75.7)	74.5 (58–77)	63.5 (53–70.5)	0.895	**0.012**	**0.011**
Post-treatment 2nd week	25.8 ± 16.9 **< 0.001 ^a^**	42.2 ± 16.4 **< 0.001 ^a^**	44.4 ± 16.7 **< 0.001 ^a^**	**0.011**	**0.008**	0.919
Post-treatment 4th week	20 (14–42) **< 0.001 ^b^**	29 (20–46) **< 0.001 ^b^**	62 (28–68.2) **< 0.001** ^b^	0.283	**0.006**	0.086
VAS						
Pre-treatment	8 (6.5–9.5)	8 (7–8)	7 (6–8)	0.739	**0.042**	**0.017**
Post-treatment 2nd week	3 (2–4) **< 0.001 ^a^**	4 (2–7) **< 0.001 ^a^**	5 (3–6.5) **< 0.001 ^a^**	0.134	**0.023**	0.547
Post-treatment 4th week	2 (0–3) **< 0.001 ^b^**	2 (1–3) **< 0.001 ^b^**	5 (2.5–6) **< 0.001 ^b^**	0.834	**0.037**	**0.035**
VAS EXT						
Pre-treatment	10 (9–10)	9 (7.5–10)	9 (8–10)	0.110	0.286	0.703
Post-treatment 2nd week	2.9 ± 2.0 **< 0.001 ^a^**	5.0 ± 2.4 **< 0.001 ^a^**	6 ± 2.6 **< 0.001 ^a^**	**0.037**	**0.005**	0.561
Post-treatment 4th week	3 (1–6) **< 0.001 ^b^**	4 (2–5) **< 0.001 ^b^**	8 (4–10) **< 0.001 ^b^**	0.552	**0.019**	**0.029**
Nirschl						
Pre-treatment	6 (5–6)	6 (5–7)	5 (4–6)	0.445	0.785	0.309
Post-treatment 2nd week	2 (1–3) **< 0.001 ^a^**	2 (1–4) **< 0.001 ^a^**	5 (4.5–6) **< 0.001 ^a^**	0.405	**0.029**	0.175
Post-treatment 4th week	2 (0–3) **< 0.001 ^b^**	2 (1–3) **< 0.001 ^b^**	5 (2.5–6) **< 0.001 ^b^**	0.834	**0.037**	**0.035**
Hand Grip						
Pre-treatment	25.5 ± 11.3	19.4 ± 8.3	28.1 ± 8.6	0.126	0.723	**0.022**
Post-treatment 2nd week	32.1 ± 11.1 **< 0.001 ^a^**	22.0 ± 10.6 **< 0.001 ^a^**	30.7 ± 12.1 **< 0.001 ^a^**	**0.011**	0.923	**0.040**
Post-treatment 4th week	32.0 ± 12.4 **0.035 ^b^**	23.3 ± 14.1 **0.037 ^b^**	28.7 ± 12.6 0.597 ^b^	0.133	0.951	0.266

Mean ± SD and median (IQR) values are presented. Significant *p* values are shown in bold. ^a^ The comparison between the pre-treatment period and the 2-week post-treatment period. ^b^ The comparison between the pre-treatment period and the 4-week post-treatment period.

## Data Availability

The data presented in this study are available on request from the corresponding author. The data are not publicly available due to patients’ privacy.

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
