# Peer review of "Early Results of Kinesio Taping and Steroid Injections in Elbow Lateral Epicondylitis: A Randomized, Controlled Study"

_medicina, 2021, doi:10.3390/medicina57040306_

Round 1

Reviewer 1 Report

  1. Page 2, line2 81-82, ‘The alpha level was determined as 0.005, the power as 90%, and the confidence interval as 95%.’ That seem had a typo, alpha level might be 0.05.
  2. An RCT study need the CONSORT flow diagram.
  3. Please clarify how to ‘blinding’.
  4. Table 2 lack of the pretreatment hand grip data. Why the KT group had lower grip power?
  5. All data in Table 3 can not make the conclusion ‘We believe KT is an alternative treatment option as effective as CSI because of the long-term uncertainties of CSI in the treatment of LE.’

Author Response

Dear Reviewer 1,

Thank you very much for your valuable evaluations and corrections. Authors’ response was given in attached file.

Kind Regards.

Reviewer 2 Report

Reviewer Comments:

Manuscript ID: medicina-1128469
Title: Early Results of Kinesio Taping and Steroid Injections in Elbow Lateral Epicondylitis: A Randomized, Controlled Study 

Reviewer
General comments

It is an article with regard to the use of Kinesio tapping (KT), steroid injection in elbow lateral epicondylitis. In general, the article is clearly written, however, there are several comments as below.

Specific comments -
1. Material and methods, I am uncomfortable about your definition of control group. Actually, the “wait-and-see” or “rest” could be accepted as one of the treatment methods. I would like to suggest the use of medication (NSAID) or rest group, instead of control group.

2.The author might consider to use an informal tapping, for example, not along the direction of the extensor carpi radialis brevis (ECR) and extensor digitorum communis (EDC), as a placebo group. In this way, it would be more accurate to evaluate the real impact of KT.

3.It was rough to estimate the true daily activity level before and after three interventions. Although all patients had been instructed to avoid activities that might intensely irritate the elbow, the patients with Kinesio tapping could be possible to have less activities because of its substantial immobilization around the elbow joint. The authors should provide more evidences to verify their hypothesis that the efficacy of Kinesio tapping to reduce pain by helping to reduce irritation in chemical receptors in the area where it is applied, improving circulation, and possibly in reducing pain with the door control mechanism.

Author Response

Dear Reviewer 2,

Thank you very much for your valuable evaluations and corrections. Authors’ response was given in attached file.

Kind Regards.

Reviewer 3 Report

Congratulations to your well written original article, entitled “Early Results of Kinesio Taping and Steroid Injections in Elbow Lateral Epicondylitis: A Randomized, Controlled Study. You prospectively evaluated 53 patients with lateral epicondylitis and tested for differences in functional scores in three different groups. Due to obvious differences in the treatment regime of 3 different groups no blinding was possible.

After 2 weeks no significant differences in the groups were detected. However, after 4 weeks an ongoing beneficious effect of KT was seen.

You should take care to better highlight the novelty of your study, as LE treatment with KT and steroid infiltration has already been evaluated in detail. Further, there are major and minor considerations listed below that should be integrated in your manuscript to make it suitable for publication in Journal of Clinical Medicine. I recommend that major revision is warranted.

Abstract:

  • Line 20/21: Please briefly define the treatment regime of the control group (wait and see).
  • Line 25: “(…) improvements were not as significant as they were in the first two weeks (…).”: Be careful not to mix up statistical significance with different heights of scores. Same in lines 161-162.
  • The presentation of the results in the abstract has a narrative character. Please add the most relevant findings in the score in terms of absolute numbers (score results).

Introduction:

  • Line 59: “palliative” -> conservative? Please check and eventually rephrase.
  • Line 63 “new method”. Please rephrase and state that KT treatment for LE is not a new method.

(Shakeri et al. 2017: The effects of KinesioTape on the treatment of lateral epicondylitis

Zhong et al. 2020: Kinesio tape reduces pain in patients with lateral epicondylitis: A meta-analysis of randomized controlled trials)

  • Line 64 “traditional method” please rephrase. E.g. “established method”.
  • Your Introduction should lacks a final paragraph that highlights the novelty of your work. Please explain how your study distinguishes from existing literature (e.g. see above two references) and what new knowledge is added to the body of the literature.

Methods:

  • Please describe, when and how often KT application was changed within the four weeks.
  • How did you assess patient compliance to the given treatment regime?
  • Lines 110-112 Anti-inflammatory drugs: Please provide the numbers, how many patients in the control group actually were prescribed NSAIDs. Can those patients be compared to patients without NSAID treatment? Please discuss!

Results:

  • Add a patient enrolment chart (e.g. flowchart) displaying all inclusion and exclusion criteria and loss to follow-up.
  • Please also show statistical insignificance of demographic data at least by “p>0.05” or give the absolute p-values. Consider showing them either in the text or in table 2.
  • Line 94: Why do you point out the KT was applied by two researchers? Do you mean physicians? Please check. Same in line 106.
  • Table 2 shows if the dominant hand and the affected hand were right or left. You should also give the combination of these data and show in how many cases the dominant hand was affected in the groups. It is presumably more convenient to stick to the introductions e.g. avoiding activities with the non-dominant hand than the other way around. Consequently this might be important information that could influence the outcomes.
  • Did you record, how long the symptoms of LE were on-going before of the start of treatment? This would be another valuable information to better characterize the control group. Did you assess the overall duration of symptoms? Please add this information or discuss.

Discussion

  • How do you estimate possible placebo effects of the different treatments after 4 weeks? If KT had to be freshly applied several times during the four weeks and potential initial placebo effect could have been “renewed” each time. Could this be considered a possible confounder for the better results of the KT group in the final week? Please discuss.

Author Response

Dear Reviewer 3,

Thank you very much for your valuable evaluations and corrections. Authors’ response was attached to file.

Kind Regards.

Reviewer 4 Report

This study confirmed changes in pain and function after cortisol steroid injection, KT, and no intervention (control) in lateral epicondylitis patients. However, revision is need, including research methods and procedures.

Introduction

  1. Line 40-41, reference is required.
  2. KT is generally applied for 3-5 days. You need a reference in the line 49-50. Also, a reference to the KT effect is needed in line 50-51.
  3. It is necessary to describe the purpose of this study in more detail.

Material and methods

  1. Line 85-92, Please describe in detail the procedure for selecting a research topic.
  2. What does mean the 50 patients (53 elbows), 20 patients (21 elbows), 15 patients (17 elbows) patients?
  3. Line 94, “by Kase et al. by two certified researchers (3-10)”, what does mean ‘(3-10)’?
  4. Please explain with subtitle as intervention and outcome measure.

Please explain in detail the application of KT, CSI intervention (eg, frequency, duration, etc.)

Please explain the outcome measure (eg, pain and function, etc).

Describe the reliability of the evaluation tool

Results

  1. Write the pre-treatment and post-treatment in Table 2

Discussion

  1. Compared to other studies, there is insufficient discussion about the results of this study.

Author Response

Dear Reviewer 4,

Thank you very much for your valuable evaluations and corrections. Authors’ response was given in attached file.

Kind Regards.

Round 2

Reviewer 1 Report

No further comment.

Author Response

Dear Reviewer 1,

We would like to thank you for your supportive suggestions

Kind regards

Firat Erpala M.D.

Reviewer 3 Report

Thank you for re-submitting your revised manuscript. I confirm, that you sufficiently processed all suggestions and were able to further improve the text. 

Author Response

Dear Reviewer 3,

We would like to thank you for your supportive suggestions

Kind regards

Firat Erpala M.D.

Reviewer 4 Report

  1. The frequency, duration, etc. of the intervention application are not provided.
  2.  qDASH, QDash is used in text.
  3. In table 3, make a table that is easy to understand by filling in the pre treatment.

Author Response

Dear Reviewer 4,

We would like to thank you for your supportive suggestions.

Corrections were made according to your advise.

  1. In Lines 112-115 we tried to be more specific about frequency and duration of the tape. Lines 129 and 130 were added.
  2. qDASH was replaced with QDASH in the manuscript.
  3. Pre-treatment data were added in Table 3.

Kind regards

Firat Erpala M.D.